



# THE MUMBA CAMPAIGN: MEASUREMENTS OF
# URBAN, MARINE AND BIOGENIC AIR
**Clare Paton-Walsh[1*], Élise-Andrée Guérette[1], Dagmar Kubistin[1], Ruhi Humphries[1,2],**
**Stephen R. Wilson[1], Doreena Dominick[1], Ian Galbally[1,2], Rebecca Buchholz[1,3],**
**Mahendra Bhujel[1,2], Scott Chambers[4], Min Cheng[2], Martin Cope[2], Perry Davy[5],**
**Kathryn Emmerson[2], David W. T. Griffith[1], Alan Griffiths[4], Melita Keywood[2],**
**Sarah Lawson[2], Suzie Molloy[2], Géraldine Rea[1,6], Paul Selleck[2], Xue Shi[1], Jack**
**Simmons[1] and Voltaire Velazco[1]**
[1] Centre for Atmospheric Chemistry, School of Chemistry, University of Wollongong,
Northfields Avenue, Wollongong, N.S.W. Australia.
[2] CSIRO Climate Science Centre, Aspendale, Victoria, Australia.
[3]Atmospheric Chemistry Observations & Modeling (ACOM) Laboratory, National Center for
Atmospheric Research, Boulder, CO, USA
[4] ANSTO, Environmental Research, Locked Bag 2001, Kirrawee DC, NSW 2232, Australia.
[5] GNS Science, National Isotope Centre, Lower Hutt, NZ.
[6] Université Pierre et Marie Curie, Laboratoire de Météorologie Dynamique - CNRS/IPSL
Ecole Polytechnique 91128 Palaiseau Cedex, Paris, France.
* Author to whom correspondence should be addressed; Clare Paton-Walsh (Murphy)
(E-Mail: clarem@uow.edu.au); Tel.: +61-2-4221-5065; Fax: +61-2-4221-4287;

-14, 2017


**Abstract**
The Measurements of Urban, Marine and Biogenic Air (MUMBA) campaign took place in
Wollongong, New South Wales (a small coastal city approximately 80 km south of Sydney,
Australia), from 21$^{st}$ December 2012 to 15$^{th}$ February 2013. Like many Australian cities,
Wollongong is surrounded by dense eucalyptus forest and so the urban air-shed is heavily
influenced by biogenic emissions. Instruments were deployed during MUMBA to measure
the gaseous and aerosol composition of the atmosphere with the aim of providing a detailed
characterisation of the complex environment of the ocean/forest/urban interface that could be
used to test the skill of atmospheric models. Gases measured included ozone, oxides of
nitrogen, carbon monoxide, carbon dioxide, methane and many of the most abundant volatile
organic compounds. Aerosol characterisation included total particle counts above 3 nm, total
cloud condensation nuclei counts; mass concentration, number concentration size
distribution, aerosol chemical analyses and elemental analysis.
The campaign captured varied meteorological conditions, including two extreme heat events,
providing a potentially valuable test for models of future air quality in a warmer climate.
There was also an episode when the site sampled clean marine air for many hours, providing
a useful additional measure of background concentrations of these trace gases within this
poorly sampled region of the globe. In this paper we describe the campaign, the meteorology
and the resulting observations of atmospheric composition in general terms, in order to equip
the reader with sufficient understanding of the Wollongong regional influences to use the
MUMBA datasets as a case study for testing a chemical transport model.  The data is
available from PANGAEA (see http://doi.pangaea.de/10.1594/PANGAEA.871982).
**Keywords:** VOCs, Ozone, Greenhouse Gases, Aerosols, Air Quality, Measurement
Campaign,

## 1. Introduction

The value of intensive measurement campaigns in helping to understand and characterise
local atmospheric composition and air quality has been recognised from as early as 1969,
when the Los Angeles Smog Project took place [*Whitby et al.*, 1972b]. Since then, many such
campaigns have focused on understanding the formation of photochemical smog in the most



polluted cities worldwide, with early efforts concentrated in the USA, (e.g. in [*Gray et al.*,
1986; *Husar et al.*, 1972; *Whitby et al.*, 1972a]). The formation of secondary organic aerosol
has also been of particular interest, with many studies using elemental carbon (black carbon)
as an indicator of primary emissions; when the ratio of organic carbon to elemental carbon in
the sampled air is higher than expected from the ratio of the primary emissions, secondary
organic aerosol formation is indicated [*Castro et al.*, 1999; *Gray et al.*, 1986; *Turpin and*
*Huntzicker*, 1995].
In Australia, there have been a number of studies aimed at improving our understanding of
ozone chemistry in the cleaner southern hemisphere atmosphere [*Galbally et al.*, 2000;
*Monks et al.*, 1998]; secondary aerosol formation [*Cainey et al.*, 2007] or other air quality
issues, such as air toxics and smoke [*Hinwood et al.*, 2007; *Keywood et al.*, 2015]. There have
also been some air quality studies specifically aimed at testing the Australian Air Quality
Forecasting System [*Cope et al.*, 2004] in Sydney [*Hess et al.*, 2004] and Melbourne [*Tory et*
*al.*, 2004]. The primary focus of these studies was testing the prediction of ozone levels in the
urban environment [*Cope et al.*, 2005]. More recent studies have examined regional air
quality in Wollongong [*Buchholz et al.*, 2016] and the effect of a major fire event on air
quality in Sydney and Wollongong [*Rea et al.*, 2016]. There have also been Australian
campaigns focused on understanding aerosol formation and composition, in the urban
environment e.g. [*Cheung et al.*, 2011; *Cheung et al.*, 2012]; coastal environments [*Cainey et*
*al.*, 2007; *Fletcher et al.*, 2007; *Modini et al.*, 2009] and within eucalypt forests [*Ristovski et*
*al.*, 2010; *Suni et al.*, 2008]. In addition, there have been some detailed studies to characterise
the concentrations of VOCs in the clean background atmosphere in the Australasian region
[*Colomb et al.*, 2009; *Galbally et al.*, 2007; *Lawson et al.*, 2015].
In this overview paper, we describe a measurement campaign in the small Australian coastal
city of Wollongong, of approximately 292,000 residents. The Wollongong region is bounded
by ocean to the east and by a steep escarpment, covered in eucalypt forest, to the west. The
coastal plain is roughly triangular in shape, being very narrow in the north where the
escarpment meets the sea, and roughly 20 kilometres wide in the south. The region spans
about 50 kilometres of coastline.
The MUMBA campaign involved collaboration between three Australian research groups
(the University of Wollongong; the Commonwealth Scientific and Industrial Research
Organisation (CSIRO), and the Australian Nuclear Science and Technology Organisation



(ANSTO), and one research organisation from New Zealand (GNS Science). MUMBA was
designed to provide a comprehensive characterisation of the local atmosphere that could test
the capabilities of air quality models to forecast atmospheric composition. Influences from
the nearby ocean sources, urban emissions and the biogenic emissions from the surrounding
eucalypt forests were expected to impact the site. This campaign aimed to make detailed
measurements of atmospheric composition under the combined influence of these different
sources, all of which typically affect the populated regions of the East coast of Australia.

## 2. Measurement Sites

The MUMBA campaign included instruments that were run at several different, nearby sites.
The main measurement site (34.397°S, 150.900°E) of the MUMBA campaign was located in
a suburban area of Wollongong approximately half a kilometre from the ocean. The
instruments were located in and adjacent to an unused hut located at the University of
Wollongong's campus east (see Figure 1a). Most instruments sampled from a mast at a height
of ~10 m above the surrounding ground level (also shown in Figure 1a). Immediately
surrounding the measurement site is a grassy plain with a suburban road to the east and a strip
of forested parkland beyond, before the sand dunes and ocean. Prevailing easterly sea breezes
brought air-masses from the ocean to the site during the day. Urban influences from the local
metropolitan area and a large industrial area, including a steelworks, typically occurred in
still conditions or with southerly winds. The steep forested escarpment is about 3 km directly
to the west of the site and approximately 400 m high, with the area beyond dominated by
eucalypt forest, such that westerly winds brought strong biogenic signals. The population
density within the surrounding area of New South Wales (NSW), including Wollongong and
Sydney is shown in Figure 1b.
The locations of different measurement sites are shown in Figure 1c.

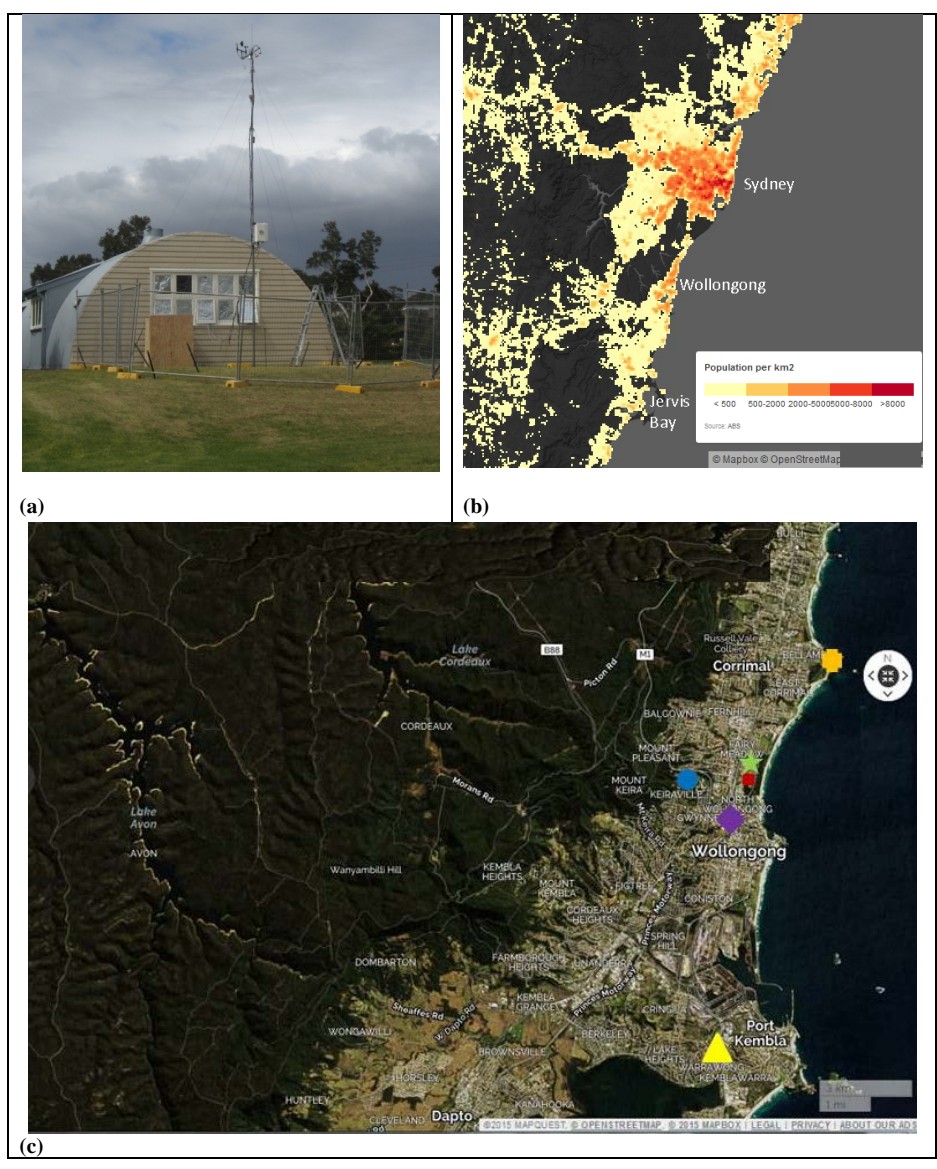

**Figure 1: (a) Hut that hosted most of the instruments during MUMBA and the sample mast. (b) Population density map for the region based on Australian Bureau of Statistics data from August 2011 – http://www.abs.gov.au/AUSSTATS/abs@.nsf/Lookup/1270.0.55.007Main+Features12011?OpenDocument) (c) Satellite view of the region showing the main MUMBA site (green star), the Wollongong Science Centre (red square), Wollongong EPA Air Quality station (purple diamond), the University of Wollongong (blue circle), Bellambi Automatic Weather Station (orange hexagon) and the ANSTO radon detector site at Warrawong (yellow triangle). The large red square indicates the approximate location of the CHIMERE grid-space used to compare to the MUMBA observations. Also visible is the large industrial area at Port Kembla and the extensive forested regions to the West. The image was created using website: www.mapquest.com "© OpenStreetMap contributors".**



In addition, ANSTO provided measurements of atmospheric radon concentrations from
Warrawong (34.48°S, 150.89°E), an industrial suburban site located approximately 10 km to
the south of the main MUMBA site. The use of radon to characterise boundary layer mixing
[*Chambers et al.*, 2011] is likely to be especially useful for testing air quality models, due to
the challenges of modelling within the complex topography of coastal areas. The locations of
all of the sites used in the MUMBA campaign are marked on the satellite view of the region
shown in Figure 1c.
**3. Description of the Instruments Deployed at the Main Measurement**
**Site**
A large range of instrumentation was deployed to enable a detailed characterisation of
atmospheric composition during the campaign. All measurements made during the campaign
are listed in Table 1. As not all instruments operated for the entire campaign, the dates of
operation of each instrument are given. All data are available from PANGAEA
(https://doi.pangaea.de/10.1594/PANGAEA.871982) as hourly averages unless otherwise
specified. Further details of the instruments are given in the Appendix, along with a second
table that lists the specific VOCs measured during the campaign and their limits of detection.
Also available from PANGAEA are the radon measurements made at Warrawong by ANSTO
and the air quality data from the Wollongong Office of Environment and Heritage (OEH)
station. The FTIR spectrometer uses a drier on the inlet and measured mole fraction in dry
air; other gas phase instruments measured in ambient air. All times are reported in local
standard time (UTC +10).
**4. Additional Measurements**
Measurements were also available from the OEH air quality station at Wollongong
(34.419°S, 150.886°E). Additional instruments were operated nearby at the University of
Wollongong's main campus (at 34.406°S, 150.897°E) [*Buchholz et al.*, 2016] and at the
nearby Science Centre (34.401°S, 150.900°E), but the observations are not included here.
Data from the University of Wollongong include retrievals of total column amounts of trace
gases from the Total Carbon Column Observing Network (see http://www.tccon.caltech.edu/)
and the Network for Detection of Atmospheric Composition Change (see
http://www.ndsc.ncep.noaa.gov/) and in situ greenhouse gas measurements (see
http://doi.pangaea.de/10.1594/PANGAEA.848263). The instrument installed at the Science





Centre was a Multi-Axis Differential Optical Absorption Spectrometer, and the data is
available from the authors upon request. The Australian Bureau of Meteorology operates an
Automatic Weather Station (AWS) at Bellambi (34.37°S, 150.93°E). Again, the data are not
included here but can be requested from the Bureau if needed.





2

**Table 1: Measurements made during the MUMBA campaign, tabulated alongside the time resolution, the instrument type, and the dates the instrument was operational. Instruments which ran for the full 8 weeks of the MUMBA campaign are shaded in dark grey, aerosol instruments that ran for the second half of the campaign are shaded in light grey and instruments that ran for a shorter time period have a white background.**

| Measured Parameter(s) | Instrument/Technique | Measurement Time Resolution | Reported Time Resolution | Reported units | Measurement Period |
|---|---|---|---|---|---|
| $O_3$ | UV (Thermo 49i) | 1 min | 1 hr | ppb | Dec 21st – Feb 15th |
| NO NO$_2$+ | Chemiluminescence, (Thermo 42i) molybdenum converter | 1 min | 1 hr | ppb | Dec 21st – Feb 15th |
| VOCs | PTR-MS (Ionicon) | ~3 min | 1 hr | ppb | Dec 21st – Feb 15th |
| $CO_2$ CO, $CH_4$, $N_2O$ del $^{13}C$ in $CO_2$ | FTIR in situ analyser | ~3 min | 1 hr | ppm ppb per mille | Dec 21st – Feb 15th |
| Boundary layer height | Elastic backscatter at 355nm - LIDAR (Leosphere ALS-400) | 30 s | 20 minutes | metres above ground level | Dec 21st – Feb 15th |
| wind speed wind direction temperature pressure relative humidity | Campbell Scientific EasyWeather | 1 min 5 min | 1 hr 1 hr | m/s degrees degree Celsius mbar % | Dec 21st – Jan 25th Jan 25th – Feb 15th |
| Total number concentration of condensation nuclei >3nm | Ultrafine Condensation Particle Counter (TSI 3776) | 1 s | 1 hr | particles/cm$^3$ | Jan 16th – Feb 15th |
| Total number concentration of cloud condensation nuclei | Cloud Condensation Nuclei Counter (Droplet Measurement Technologies) | 1 s | 1 hr | particles/cm$^3$ | Jan 16th – Feb 15th |
| Particle number size distribution (~14 nm to ~660 nm) | Scanning mobility particle sizer | ~ 5 min | 1 hr | diameter: nm particle concentration: dN/dLogDp particles/cm$^3$ | Jan 16th – Feb 15th |
| Elemental and organic carbon in PM2.5 fraction | HiVol sampling – chemical analysis | 04:00-09:00 and 10:00-18:00 daily | 04:00-09:00 and 10:00-18:00 daily | ug C/m$^3$ | Jan 16th – Feb 15th |
| PM$_{10}$ and PM$_{2.5}$ elemental | Streaker sampler (PIXE) – ion beam analysis | 1 hr | 1 hr | ng/m$^3$ | Jan 21st – Feb 15th |



| composition | | | | | |
|---|---|---|---|---|---|
| PM$_{2.5}$ mass concentration | Laser scattering (Met One eSampler) | 5 min | ug/m$^3$ | 1 hr | Jan 24$^{th}$ – Feb 15$^{th}$ |
| NO, NO$_2$ | Chemiluminescence, blue light converter | 1 min | ppb | 1 hr | Feb 1$^{st}$ – Feb 15$^{th}$ |
| carbonyls and ketones | 2,4-DNPH cartridges/high performance liquid chromatography | 04:00-09:00, 10:00-18:00 and 18:00 -04:00 daily | ppb | 04:00-09:00, 10:00-18:00 and 18:00 - 04:00 daily | Feb 4$^{th}$ – Feb 15$^{th}$ |



## 5. Meteorology during the MUMBA Campaign

The summer of 2012 - 2013 was the hottest summer on record for Australia at the time [*White* and *Fox-Hughes*, 2013]. There were two extremely hot days in the Wollongong region during MUMBA, with maximum temperatures of 40.4 °C on January 8th and 42.4°C on January 18th 2013 recorded at Bellambi AWS (both below the record of 43.7°C set on January 1st, 2006). The campaign encompassed the wettest January day on record for the region, with 139 mm of rain falling at Bellambi AWS between 08:00 on January 28th and 08:00 on January 29th 2013 (see the top panel of Figure 2).

The lower panel of Figure 2 shows the mean hourly temperature recorded from the 10 m mast at the MUMBA site over the campaign. The two extremely hot days can be clearly seen in this figure. The mean daily maximum temperatures during January 2013 was 25.7°C, which is 0.9°C above the long-term average of 24.8°C and in the 95th percentile of monthly mean maximum temperatures for January at Bellambi AWS (using data from 1988 to the present day).

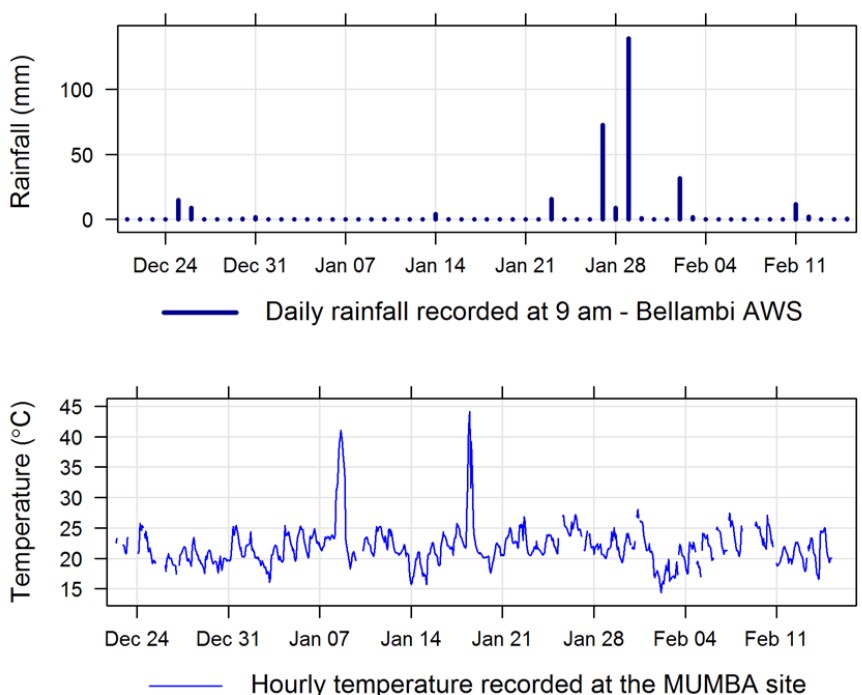

**Figure 2: Upper panel shows a bar chart of daily rainfall in millimeters from Bellambi AWS. Lower panel shows the time-series of mean hourly temperature measured during MUMBA.**





The average wind speed recorded at the MUMBA site during the campaign was 2.8 ms[-1], and
the maximum hourly-averaged wind speed recorded was 9.2 ms[-1]. The 1[st], 2[nd] (median) and
3[rd] quartiles of the wind speed were 1.4 ms[-1], 2.6 ms[-1] and 3.9 ms[-1] respectively. Figure 3
shows the composite diel cycles of wind speed and wind direction as measured at the main
MUMBA site. The general pattern was of a relatively strong sea breeze during the day
(~easterly winds of 3-4 ms[-1]) and of calmer conditions overnight. Westerly winds were more
frequent during night time (although north-easterly winds sometimes persisted into the night).
This pattern was repeated all over the local region (as shown in data from OEH air quality
sites and from the University of Wollongong) [*Guérette*, 2016].

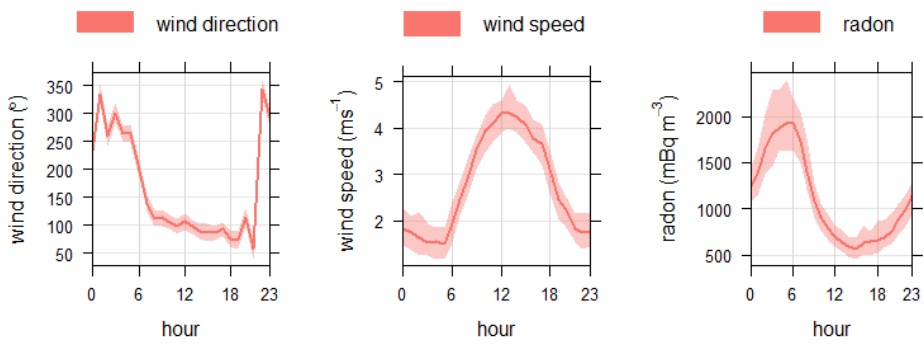

**Figure 3: Observed diurnal cycles of wind direction and wind speed at the main MUMBA site and radon**
**concentration at Warrawong observed during the campaign. (Shaded area shows 95% confidence**
**interval from a bootstrap resampling of the data. See [*Carslaw and Ropkins*, 2012] for description of this**
**and of calculations of average wind direction).**
The third panel in Figure 3 shows the composite diel cycle of radon measured at the ANSTO
site in Warrawong. The radon plot shows a build up at night with a peak in the early hours of
the morning, indicating a shallower and more stable boundary layer at night than during the
day, with the boundary layer at its shallowest around 05:00 or 06:00 AEST. During the day,
due to heating at the surface and other processes, the boundary layer grows deeper and more
turbulent; this is reflected in the lower radon values observed during the day. Minimum radon
levels in the afternoon are also influenced by the fetch of the air reaching the site, with air
that has travelled over the ocean containing less radon than air that has travelled over land
[*Chambers et al.*, 2015]. In the Wollongong region, an increased boundary layer height and
strong sea breezes combine to produce the low radon levels observed in the afternoon.
Comparisons of the winds measured at the MUMBA site during the campaign, to
simultaneous measurements at the three air quality sites operated by the Office of
Environment and Heritage in the area (at Wollongong, Kembla Grange, and Albion Park),





indicated that the wind patterns observed at the MUMBA site were generally representative
of the region as a whole [*Guérette*, 2016]. Long-term average wind data at 15:00 each day are
publically available from the Bellambi AWS from 1997 – 2010, and this was used for
comparison with the wind data recorded at this time throughout January during the campaign.
The MUMBA site in January 2013 was characterised by slightly less frequent northerly
winds and slightly more frequent westerly winds than expected from the long-term average at
Bellambi, but otherwise wind patterns were very similar in the two records. The MUMBA
site experienced lower wind speeds than the long-term averages at Bellambi (but this may be
due to location differences rather than atypical weather patterns) [*Guérette*, 2016]. Thus we
conclude that the measurements made at the MUMBA site during the campaign should be
broadly representative of the region, as well as of the summer season.

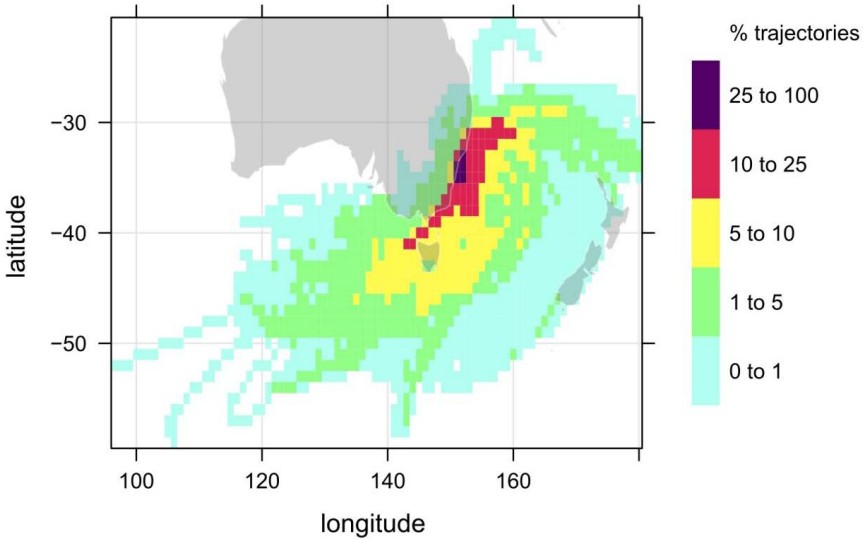

**Figure 4: 96 hour gridded back trajectory frequencies during MUMBA. The surface is coloured by the**
14        **percentage of total trajectories which pass through each grid-box.**

On a larger scale, the dominant circulation pattern during MUMBA was anti-cyclonic, with
the main fetch being principally oceanic (as opposed to continental), which is typical of
summer [*Chambers et al.*, 2011]. This is illustrated in Figure 4, which shows a gridded back
trajectory frequency plot for 96-hour pre-calculated back trajectories made available for
Wollongong through the Openair package [*Carslaw and Ropkins*, 2012]. The trajectories
were calculated using the HYSPLIT trajectory model (Hybrid Single Particle Lagrangian
Integrated Trajectory Model; http://ready.arl.noaa.gov/HYSPLIT.php) every three hours,



from an initial height of 10 metres and propagated backwards in time for 96 hours using the
Global NOAA-NCEP/NCAR reanalysis meteorological fields at 2.5° horizontal resolution.
The surface of the plot is coloured by the percentage of total trajectories which pass through
each grid-box.

## 6. Urban, Marine and Biogenic Influences during the MUMBA Campaign

The MUMBA campaign was designed to characterise atmospheric composition at the
ocean/forest/urban interface and thereby provide a dataset that could be used to test the skill
of atmospheric models within a coastal environment. In this section, the major urban, marine
and biogenic sources that influence atmospheric composition in the region are described.
The dominant anthropogenic sources in the region are the Port Kembla steelworks, located
approximately 10 km south of the main MUMBA site (for $PM_{2.5}$, $PM_{10}$, CO, $NO_X$ and $SO_2$)
and motor vehicles (for $NO_X$, CO and VOCs) (see: http://www.npi.gov.au/npidata). The
ocean lies to the east of the site and large forested areas to the west. Outflow from the Sydney
basin (80 km to the north) may accompany winds from the north-east.
The impact of the different air-masses sampled can be illustrated using a bivariate polar plot,
which shows how a pollutant varies by wind speed and wind direction as suggested by
*Carslaw et al* [2006]. Figure 5a shows a bivariate polar plot for CO measured from the main
MUMBA site throughout the campaign. Several distinct regions are evident, with the most
obvious being the very high amounts of CO that are measured when the site experiences
southerly winds with speeds between 2 and 6 ms$^{-1}$. This direction brings air-masses over
central Wollongong and also over the industrial area centred on the steelworks at Port
Kembla. In contrast, easterly to south-south-easterly winds bring very low amounts of CO to
the MUMBA site as the air-masses come from the Pacific Ocean. There were a number of
occasions during the campaign when easterly winds brought predominantly marine air to the
measurement site. These periods were identified by using radon values below a threshold of
200 mBq m$^{-3}$, indicating minimal terrestrial impact in agreement with back trajectories. One
episode in particular, on December 26$^{th}$, 2012, lasted several hours and was characterised by
greenhouse gas concentrations similar to those measured in December 2012 at the Cape Grim
baseline air pollution station on the north west tip of Tasmania, Australia (40.683°S, 144.689°
E) (see http://www.csiro.au/greenhouse-gases/). These episodes are explored further in a
paper dedicated to the marine air signature during MUMBA [*Guérette et al.*, 2017].



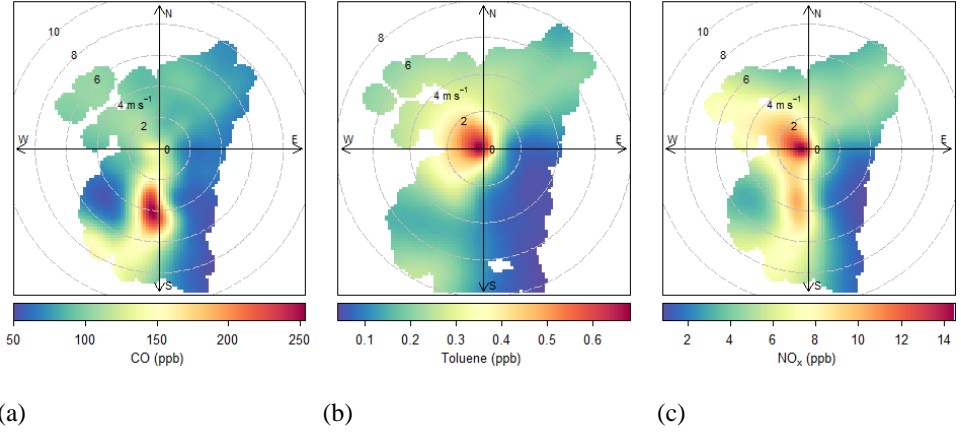

(a)                                    (b)                                    (c)

**Figure 5: Bivariate polar plots showing how mole fractions (ppb) of (a) CO, (b) toluene and (c) NO$_X$. vary as a function of wind speed (ms$^{-1}$) and wind direction at the main MUMBA site during the campaign. Wind speed is represented by the concentric circles and wind direction is shown as compass directions, such that the shape of the coloured area illustrates the wind speeds and directions experienced during the campaign. The colour indicates mean mole fraction measured under the corresponding wind conditions.**

CO mole fractions from the north-east (that also come off the ocean) are nearly double those from the south-south-east, indicating that the MUMBA site may be influenced by outflow from the Sydney basin, 80 km to the north. Elevated CO is also measured from the north-west in the direction of the nearest suburban shopping centre, multilane road and local industrial sites (including a coke-works and mining operations). In contrast, relatively low concentrations are seen from the south-west where there is a steep escarpment and eucalypt forests beyond.

Figure 5b shows the polar bivariate plot for toluene, which is predominantly emitted from motor vehicles and is not emitted from the steelworks. The plot shows the largest concentrations with low wind speeds, as is indicative of local sources building up in the nocturnal boundary layer, however there is a directional bias with much cleaner air to the east. This is due to clean marine air coming from the east and is also obvious in the low amounts of toluene coming from all wind speeds from the south-east. In contrast, there are slightly higher mole fractions of toluene that accompany winds from the north-east, again indicating possible outflow from Sydney or more local pollution to the north that is brought in on the sea breeze.

Figure 5c shows the polar bivariate plot for NO$_X$, which shows a mixture of the features seen in the toluene and CO plots, indicative of a mixture of traffic and industrial sources as expected.



1. In Figure 6 polar bivariate plots are shown for the main criteria pollutants of concern within the air-shed (PM$_{2.5}$ and O$_3$), along with the most significant biogenic volatile organic compounds, isoprene (PTR-MS m/z 69) and monoterpenes (PTR-MS m/z 137). Both isoprene and monoterpenes show very elevated concentrations with strong north-westerlies, which occurred on the two extremely hot days (January 8$^{th}$ and January 18$^{th}$ 2013). The monoterpenes are also high with still winds, because (unlike isoprene) these compounds are also emitted during the night and hence build up in the nocturnal boundary layer. Also, under more stable night-time conditions, katabatic flow down the escarpment will bring air predominantly influenced by the eucalypt forests to the site.

The highest PM$_{2.5}$ concentrations are seen with strong to moderate winds from the south, which bring industrial sources from the Port Kembla steelworks. Elevated PM$_{2.5}$ is also seen with north-westerly winds that bring biogenic influences from the escarpment and densely forested regions beyond. Highest O$_3$ concentrations are also seen with the hot north-westerly winds, with the influence of NO$_X$ titrating out the O$_3$ clearly seen with low concentrations observed at low wind speeds and with wind from the south. The high O$_3$ and PM$_{2.5}$ values that accompany the high levels of isoprene and monoterpenes, imply that biogenic influences are important for both O$_3$ formation and secondary organic aerosol formation in the region. This may be due to having a VOC-limited environment (the formaldehyde to NO$_X$ ratio averaged 0.3 over the campaign), coupled to the fact that anthropogenic emissions of VOCs are relatively low in the area, so that biogenic VOCs are extremely important to the overall budget. Despite the importance to air quality, biogenic emissions from Australian eucalypt forests are poorly understood [*Emmerson et al.*, 2016] and further research is needed to better characterise biogenic emissions in this region of Australia.





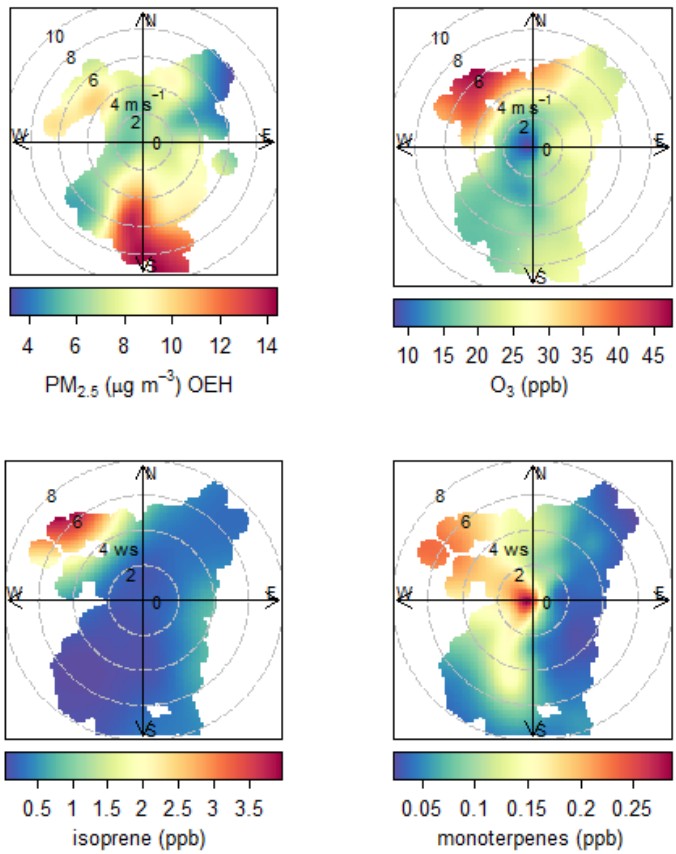

**Figure 6: Bivariate polar plots showing how (a) the concentration of PM$_{2.5}$ (µgm$^{-3}$) at the OEH station and mole fractions (ppb) of (b) O$_3$, (c) isoprene and (d) monoterpenes at the main MUMBA site, varied as a function of wind speed (ms$^{-1}$) and wind direction during the campaign.**

## 7. Summary and Conclusions

The combined datasets from MUMBA provide a useful case study for testing the skill of air quality models in the complex environment of urban, marine and forest influences that exists in coastal Australia, where the majority of its inhabitants live. This overview paper aims to provide the reader with sufficient understanding of the MUMBA campaign to use the datasets as a test case for any air quality model, including an understanding of the Wollongong urban air-shed, regional topography, emissions and meteorology.



During the eight week campaign the MUMBA site experienced some very different
conditions, ranging from relatively polluted air (with local urban pollution from traffic and
nearby industrial sources), to unpolluted marine air with composition akin to that
representative of the remote marine boundary layer measured at the Cape Grim station under
baseline conditions. There were two extreme heat events during MUMBA when westerly
winds brought strong biogenic influences from nearby forested regions. The measurements of
atmospheric composition during these events provide data that could prove to be a valuable
test of models of future air quality in a changing climate.
A series of papers are in preparation that describe the main scientific findings from the
MUMBA campaign, including articles focusing on: (1) drivers of urban air quality; (2)
marine air at 34ºS; (3) biogenic emissions of volatile organic compounds; (4) drivers of
aerosol loading in the airshed and (5) new particle formation events. In addition, the
MUMBA campaign measurements are being used in conjunction with long-term
measurements from the OEH air quality network, and campaign data from the Sydney
Particle Study campaigns [*Keywood et al.*, 2016a; *Keywood et al.*, 2016b] as observational
datasets in a modelling inter-comparison exercise involving four different regional air quality
models.     The     MUMBA     data     is     available     from     PANGAEA
(https://doi.pangaea.de/10.1594/PANGAEA.871982) for other researchers wanting to join the
inter-comparison exercise or use the data independently to test atmospheric composition
simulations in the region.
**Acknowledgements**
The authors would like to thank all those from the University of Wollongong's Centre for
Atmospheric Chemistry and CSIRO's Climate Science Centre group, who helped with the
logistics of undertaking an extensive measurement campaign, and in particular Travis Naylor,
Graham Kettlewell, Christopher Caldow, Frances Phillips, Jason Ward, James Harnwell and
Jenny Fisher. The ANSTO technical staff responsible for installing and maintaining the
Warrawong radon detector were Ot Sisoutham and Sylvester Werczynski. Thanks are also
due to Kids Uni & the Science Centre for their helpful support, and to David Carslaw (& all
the statisticians who developed the relevant 'R' code) for public access to the excellent



Openair package for analysis of air quality data. We acknowledge funding from the
Australian Research Council (for funding the campaign as part of the Discovery Project
DP110101948) and the Clean Air and Urban Landscapes hub of Australia's National
Environmental Science Programme (for funding for later analysis of results that was required
for producing this manuscript). This research was also supported by Australian Government
Research Training Program (RTP) Scholarships.

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



# Appendix 1: Details of the Instruments Used

## 1. PTR-MS

An "IONICON" proton transfer reaction mass spectrometer (PTR-MS) from CSIRO operated throughout the MUMBA campaign. The PTR-MS was installed along with the auxiliary equipment that controls the flow rate and incorporates regular sampling of calibration gases and "zero air" [*Galbally et al.*, 2007]. The instrument performed zero measurements twice daily for 40 minutes each time (at 00:50 and at 15:00 local time) by sampling ambient air that had been stripped of volatile organic compounds (VOCs) by passing through a platinum-coated glass wool catalyst heated to 350°C. A multi-species, single-point calibration was performed daily (from 01:30 until 03:00 local time) by introducing a known flow of calibration standard into the zero air stream. Calibration mole fractions were ~10 to 20 ppb for each VOC present in the standard.

The PTR-MS was operated using $H_3O^+$ ions only and was programmed to scan through its range of mass-to-charge ratios (m/z) with a dwell time of one second, for a total cycle time of about three minutes. Mole fractions of volatile organic compounds were calculated from the PTR-MS at the following masses: formaldehyde (mass 31), methanol (mass 33), acetonitrile (mass 42), acetaldehyde (mass 45), acetone (mass 59), isoprene (mass 69); isoprene oxidation products methacrolein and methyl vinyl ketone (mass 71); benzene (mass 79), toluene (mass 93), xylenes (mass 107), trimethyl benzenes (mass 121) and monoterpenes (mass 137). Further details of these measurements, calibrations and corrections can be found in *Guérette* [2016].

## 2. VOC sequencer

From 4[th] to 15[th] February 2013, continuous VOC measurements made using the PTR-MS were supplemented by integrated measurements collected on the VOC sequencer. The VOC sequencer passes air samples through two different adsorbent tubes to collect the VOCs and the carbonyls respectively. These tubes were analaysed at CSIRO on a gas chromatography flame ionisation detection/mass spectrometer (GC-FID-MS) for VOCs [*Cheng*, 2015] and HPLC for carbonyls [*Lawson et al.*, 2015], which enables unambiguous species identification (which is not always provided by product ion mass numbers from the PTR-MS) at 5, 8 or 10-hour temporal resolution [*Dunne et al.*, 2017]. Unfortunately, there was a suspected leak on the VOC tube side, (with very low concentrations measured), such that none of these data could be used. In addition there were condensation issues for the carbonyl tubes and only a



subset of the species could be determined with confidence. A list of the species measured
successfully using the sequencer is given in Table 2.

### 3.  Fourier transform Infra-red (FTIR) Trace Gas Analysers

FTIR trace gas analysers measure carbon dioxide ($CO_2$), methane ($CH_4$), carbon monoxide
(CO) and nitrous oxide ($N_2O$) in air with precision and accuracy that meet the World
Meteorological Organisation - Global Atmosphere Watch standards for baseline air. In
addition the instrument can measure and $^{13}C$ in $CO_2$ and retains the spectra allowing post
analysis for other infrared active  trace gases in highly polluted episodes [*Griffith et al.*,
2012]. The instrument ran throughout the whole MUMBA campaign, with the only data
interruption due to the cell temperature going above the range calibrated for on the 18[th]
January 2013. In theory the instrument could be retrospectively recalibrated at the higher
temperatures but since all other instruments had been switched off in the heat this was not
attempted. In addition to the instrument at the main MUMBA site, another FTIR trace gas
analyser was operated throughout the campaign at the main campus of the University of
Wollongong [*Buchholz et al.*, 2016].

### 4.  NO$_X$ and O$_3$ Monitors

Throughout the MUMBA campaign $O_3$ and $NO_X$ measurements were made using monitors
that utilised UV absorption and chemiluminescence techniques respectively. The NO-$NO_2$-
$NO_X$ monitor (Thermo Scientific Instruments, model TSI 42i,) detects NO using the
chemiluminescence technique. $NO_2$ is measured via decomposition to NO by passing over a
molybdenum converter. The difference between the NO concentrations in the two samples is
used to calculate the $NO_2$ concentration. One issue with this technique is that other nitrates
(such as PAN and $HNO_3$) may be present and are also converted to NO by molybdenum but
with different unknown efficiencies [*Steinbacher et al.*, 2007]. In order to get an indication of
the likely level of this problem a second $NO_X$ monitor from CSIRO was deployed in the last
two weeks of the campaign. This $NO_X$ monitor uses a blue-light converter so that only the
$NO_2$ is converted photolytically to NO [*Fehsenfeld et al.*, 1990]. The analysers were within
5% of each other for both NO and $NO_2$.





## 5. Micro-physical Particle Counters

From 16th January to 15th February 2013 a suite of microphysical particle counters was operated at the main MUMBA site taking ambient air through an 8 m copper inlet mounted on the mast at a height of ~9.5 m above the surrounding flat area.

An ultrafine condensation particle counter (uCPC, TSI model 3776) measured the total in-situ number concentration of condensation nuclei >3 nm. Particles enter a supersaturated butanol chamber and all particles > 3nm are grown to sizes that were able to be counted with a standard optical counter.

A Cloud Condensation Nuclei Counter (CCNC) made by Droplet Measurement Technologies was used to measure the total number concentration of Cloud Condensation Nuclei (CCN). The instrument operates by similar principle as the CPC, where aerosols are passed through a supersaturated chamber of liquid, except that water is used instead of butanol. Only particles able to act as CCN are thus activated and counted. The instrument was setup to measure particles activated at a supersaturation of 0.5%.

The particle number size distribution from ~14 nm to ~660 nm was measured with a scanning mobility particle sizer (SMPS). The SMPS (TSI model 3080 with DMA 3081 and TSI CPC 3772) ionises particles using radiation from Kr-85 decay. The charged particles then enter an electrostatic column which ramps its voltage to continually select particles based on their charge-mass ratio. Selected particles are then counted by a standard CPC.

Total $PM_{2.5}$ aerosol mass concentration measurements were also made using a Met One eSampler utilising laser scattering techniques (from 24th January to 15th February). The aerosol mass concentration is calibrated via the mass of an integrated sample collected on a filter that was changed weekly.

## 6. Filter Samplers

Filter samples of total $PM_{2.5}$ aerosol were collected twice daily using an Ecotech High Volume Air Sampler (HiVol). Integrated morning samples were collected on filters from 04:00 and 09:00 each day, with integrated afternoon samples from 10:00 to 18:00 each day. Thus two filter changes were required (one between 09:00 and 10:00 and another after 18:00 and before 04:00). The filters were taken back to CSIRO for aerosol chemical composition analysis.

A small section (~0.5 cm$^2$) of each filter was punched out and the total collected $PM_{2.5}$ aerosol analysed for its total carbon content, elemental carbon (EC) and organic carbon (OC)



content using a Thermal Optical Carbon Analyser (Model 2001A). The HiVol instrument
logs the total flow of air that has been passed through each filter and so the total carbon, EC
and OC in the integrated sample of air can be calculated in $\mu g/m^3$.
Also deployed was a Streaker Sampler from GNS Science. This sampler slowly rotates a disk
holding two filters taking ~48 hours for a full revolution. The filters were changed every two
days between 09:00 and 10:00. Only a small section of the filter is required for elemental
composition analysis such that hourly measurements of black carbon and all elements from
sodium to uranium on the periodic table are obtained.

### 9   7. LIDAR

Throughout the MUMBA campaign ANSTO provided a Leosphere ALS-400 cloud and
aerosol LIDAR that measures elastic backscatter at 355 nm, which is proportional to aerosol
density. By plotting the (range-corrected) backscatter against height, a vertical profile of
aerosol density is created. A negative gradient in aerosol density as represented in the vertical
profiles is indicative of a reduction in aerosol density, and therefore a candidate for the
boundary layer height. Boundary layer heights were estimated via two methods:
(1) Visually from plots of the logarithm of the range-corrected 355 nm signal against

17       height

(2) Using the "STRAT" algorithm [*Morille et al.*, 2007].
Since this technique relies on clear skies and sufficient aerosol loading to provide a strong
backscatter signal, it is not always possible to determine the boundary layer height with
confidence. Both estimates of boundary layer height with 20-minute resolution are included
in the PANGAEA dataset (https://doi.pangaea.de/10.1594/PANGAEA.871982).

### 23   8. Weather Station

Two different weather stations operated during MUMBA providing common meteorological
parameters including temperature, humidity, pressure, wind speed and direction. The switch
occurred on the 25th January when the original (borrowed) weather station was needed for
another field campaign. The Digitech system operated at 5-minute resolution and provided
wind direction as 16 quadrants only, whereas the original station (Campbell Scientific Inc.)
operated at 1-minute resolution and provided wind direction with degree resolution. Both
records are available on PANGAEA as hourly averages.



# 1    Appendix 2: List of VOCs Measured

| Species | formula | MW | Measurement technique | Time Resolution | n | DL (ppb) | n < DL |
|---|---|---|---|---|---|---|---|
| formaldehyde | $C_2HO$ | 30.03 | DNPH-derivatization/HPLC | 04:00 – 09:00 | 12 | 0.019 | |
| | | | | 10:00 – 18:00 | 11 | 0.012 | 0 |
| | | | | 18:00 – 04:00 | 9 | 0.009 | |
| | | | PTR-MS m/z 31 | hourly | 1027 | 0.205 0.105 0.186 | 23 |
| methanol | $CH_3OH$ | 32.04 | PTR-MS m/z 33 | hourly | 1027 | 0.050 0.033 0.062 | 0 |
| acetonitrile | $C_2H_3N$ | 41.05 | PTR-MS m/z 42 | hourly | 1027 | 0.002 0.001 0.002 | 0 |
| acetaldehyde | $C_2H_4O$ | 44.05 | DNPH-derivatization/HPLC | 04:00 – 09:00 | 12 | 0.018 | |
| | | | | 10:00 – 18:00 | 11 | 0.011 | 1* |
| | | | | 18:00 – 04:00 | 9 | 0.009 | |
| | | | PTR-MS m/z 45 | hourly | 1027 | 0.018 0.007 0.012 | 0 |
| Glyoxal | $C_2H_2O_2$ | 58.04 | DNPH-derivatization/HPLC | 04:00 – 09:00 | 12 | 0.011 | |
| | | | | 10:00 – 18:00 | 11 | 0.007 | 0 |
| | | | | 18:00 – 04:00 | 9 | 0.006 | |
| acetone | | | PTR-MS m/z 59 | hourly | 1027 | 0.010 0.013 0.007 | 0 |
| Propanal | $C_3H_6O$ | 58.08 | DNPH-derivatization/HPLC | 04:00 – 09:00 | 12 | 0.011 | |
| | | | | 10:00 – 18:00 | 11 | 0.007 | 4 |
| | | | | 18:00 – 04:00 | 9 | 0.006 | |
| isoprene | | | PTR-MS m/z 69 | Hourly | 1029 | 0.003 0.005 0.003 | 2 |
| sum of methacrolein and methyl vinyl ketone | $C_4H_6O$ | 70.09 | PTR-MS m/z 71 | Hourly | 1027 | 0.004 0.005 0.002 | 0 |
| methylglyoxal | $C_3H_4O_2$ | 72.02 | DNPH-derivatization/HPLC | 04:00 – 09:00 | 12 | 0.006 | |
| | | | | 10:00 – 18:00 | 11 | 0.003 | 0 |
| | | | | 18:00 – 04:00 | 9 | 0.003 | |
| benzene | | | PTR-MS m/z 79** | hourly | 1029 | 0.010 0.012 0.007 | 14 |




| toluene | | | PTR-MS m/z 93 | | | 1029 | 0.005 0.008 0.004 | 1 |
|---|---|---|---|---|---|---|---|---|
| hexanal | $C_6H_{12}O$ | 100.16 | DNPH-derivatization/HPLC | 04:00 – 09:00 | | 12 | 0.008 ppb | |
| | | | | 10:00 – 18:00 | | 11 | 0.005 ppb | 2 |
| | | | | 18:00 – 04:00 | | 9 | 0.004 ppb | |
| benzaldehyde | $C_7H_6O$ | 106.12 | DNPH-derivatization/HPLC | 04:00 – 09:00 | | 12 | 0.003 | |
| | | | | 10:00 – 18:00 | | 11 | 0.002 | 1 |
| | | | | 18:00 – 04:00 | | 9 | 0.002 | |
| sum of $C_8H_{10}$ compounds | $C_8H_{10}$ | 106.16 | PTR-MS m/z 107 | | | 1029 | 0.003 0.016 0.009 | 13 |
| sum of $C_9H_{12}$ compounds | $C_9H_{12}$ | 120.20 | PTR-MS m/z 121 | hourly | | 1029 | 0.003 0.013 0.006 | 2 |
| sum of monoterpenes | | | PTR-MS m/z 137 | hourly | | 1029 | 0.007 0.016 0.007 | 29 |

#note that the PTRMS was run under three differing instrumental settings (due to an accidental change in dwell time setting). Thus 3 different detection limits are listed. See metadata in PANGAEA for more details.

*An additional 11 data points were excluded due to analytical problems