# Peer review of "THE MUMBA CAMPAIGN: MEASUREMENTS OF"

_Earth System Science Data, 2017_

## Referee Comment (RC1) · Anonymous Referee #1 · 20 Mar 2017

I read this manuscript with fascination, and have no worry about the publication. The authors, and collaborators, have developed an innovative monitoring campaign. It is obvious that the paper was prepared by authors experienced in writing this type of article.

The manuscript describes the MUMBA campaign (e.g. context, background, technological design) designed to characterize atmospheric composition in a complex environment. This manuscript reports novel approaches and the research direction. The datasets provide i) a useful case study for validation of coupled models (air quality and meteorology) in a coastal environment and ii) a basis for scientific papers (in preparation).
* * *

---

## Referee Comment (RC2) · Anonymous Referee #2 · 4 Apr 2017

THE MUMBA CAMPAIGN: MEASUREMENTS OF URBAN, MARINE AND BIOGENIC AIR by Paton-Walsh et al.

The manuscript submitted by Paton-Walsh et al. describes a dataset for MUMBA campaign, which took place in a small coastal city in Australia, with the goal to provide information on atmospheric composition changes under the influences of marine air or urban and biogenic emissions.The dataset contains time series related to particles, atmospheric trace gases, speciated VOCs, radon and meteorological parameters. What makes it most interesting is that campaign captured two extreme heat events and a period when the site was under the influence of clean marine air. The provided dataset can be used for testing the chemical transport models, but not only. Present paper also introduces the future ones focused on specific issues mentioned in the Summary and Conclusion section.

[Figure]

The manuscript is logically structured and provides a good overview of the available information gathered during the measurement campaign. It is a well-written data description paper. Abstract is a good representation of the main text, the instrumentation is presented in necessary detail, the experimental methods are currently used in the field, enough explained and with references when it was necessary. The accuracy of the resulting dataset appears to be adequate. The data files are indeed public, easily accessible (under Creative Commons Attribution License) in many formats and the Summary and Conclusion section of the manuscript mentions clearly that MUMBA data is free to the scientific community.

I recommend publication of this work, subject to the very few minor comments:

1. The authors should give more details on pollution sources in the metropolitan area in Section 2.

2. The authors state their MUMBA campaign period runs from 21st December 2012 to 15th February 2013. However, the dataset on PM2.5 indicates data from 24-25 January to 15 February; carbonyls - only in February 2013; CN/CCN – from 16/15 January to 15 February; carbon fractions – from 22 January to 15 February... Although the authors indicate this in Table 1 and make a short statement in Section 3, it is not easy for reader to get a clear image of the different periods associated to the various measured parameter. I think a more elaborate explanation of these discrepancies should be added in Section 3. There are some negative values in the data set that should be also explained.

3. Page 6, line 21: please, replace "all times are reported in. . ." with "all measurements are reported in. . ."

4. Page 11, line 4 and line 15: please, replace "diel cycles" with "diurnal cycles".

---

## Author Comment (AC1) · 3 May 2017

MUMBA Overview - response to reviewers

(1) Comments from Referees

Anonymous Referee #1

I read this manuscript with fascination, and have no worry about the publication. The authors, and collaborators, have developed an innovative monitoring campaign. It is obvious that the paper was prepared by authors experienced in writing this type of article.

The manuscript describes the MUMBA campaign (e.g. context, background, techno-logical design) designed to characterize atmospheric composition in a complex environment. This manuscript reports novel approaches and the research direction. The datasets provide i) a useful case study for validation of coupled models (air quality and meteorology) in a coastal environment and ii) a basis for scientific papers (in preparation).

Anonymous Referee #2

The manuscript submitted by Paton-Walsh et al. describes a dataset for MUMBA campaign, which took place in a small coastal city in Australia, with the goal to provide information on atmospheric composition changes under the influences of marine air or urban and biogenic emissions. The dataset contains time series related to particles, atmospheric trace gases, speciated VOCs, radon and meteorological parameters. What makes it most interesting is that campaign captured two extreme heat events and a period when the site was under the influence of clean marine air. The provided dataset can be used for testing the chemical transport models, but not only. Present paper also introduces the future ones focused on specific issues mentioned in the Summary and Conclusion section. The manuscript is logically structured and provides a good overview of the available information gathered during the measurement campaign. It is a well-written data description paper. Abstract is a good representation of the main text, the instrumentation is presented in necessary detail, the experimental methods are currently used in the field, enough explained and with references when it was necessary. The accuracy of the resulting dataset appears to be adequate. The data files are indeed public, easily accessible (under Creative Commons Attribution License) in many formats and the Summary and Conclusion section of the manuscript mentions clearly that MUMBA data is free to the scientific community.

I recommend publication of this work, subject to the very few minor comments:

1.) The authors should give more details on pollution sources in the metropolitan area in Section 2.

2.) The authors state their MUMBA campaign period runs from 21st December 2012 to

15th February 2013. However, the dataset on PM2.5 indicates data from 24-25 January to 15 February; carbonyls - only in February 2013; CN/CCN – from 16/15 January to 15 February; carbon fractions – from 22 January to 15 February... Although the authors indicate this in Table 1 and make a short statement in Section 3, it is not easy for reader to get a clear image of the different periods associated to the various measured parameter. I think a more elaborate explanation of these discrepancies should be added in Section 3. There are some negative values in the data set that should be also explained.

3.) Page 6, line 21: please, replace "all times are reported in: : :" with "all measurements are reported in: : :"

4.) Page 11, line 4 and line 15: please, replace "diel cycles" with "diurnal cycles".

(2) Author's response and (3) Author's changes to manuscript.

The authors would like to thank the reviewers for their time and positive comments on the manuscript.

The specific points raised by reviewer 2 are addressed below:

1.) Agreed – the following sentence has been added at page 4, line 19: "The steelworks and surrounding industry is a large source of PM2.5 and CO, whilst traffic dominates the remainder of the urban pollution sources."

2.) Agreed.

a) Campaign periods: We tried to make this clear in Table 1 but have tried to further clarify this in the main text (page 6, line 12), which now reads: " All measurements made during the campaign are listed in Table 1, along with the dates of operation for each instrument. MUMBA operated in two distinct stages, with most gas-phase and meteorological measurements running throughout the 8 week campaign, and aerosol-phase measurements added in the second half of the campaign. A few instruments operated for different time periods and these are distinguished in Table 1 by different

background shading."

b) Negative values: The following sentences were added: "Note that some instruments can produce negative values when the concentrations are close to the detection limit. Negative concentration values (although non-physical) have not been removed from the MUMBA dataset because they are indicative of the instruments' true performance and removing negative values will produce small positive biases in calculations of longer term average concentrations.":

3.) Agreed - Done

4.) Agreed - Done

Please also note the supplement to this comment:
http://www.earth-syst-sci-data-discuss.net/essd-2017-14/essd-2017-14-AC1-supplement.pdf

─────────────────────────────